# Modifications of Chest CT Body Composition Parameters at Three and Six Months after Severe COVID-19 Pneumonia: A Retrospective Cohort Study

**DOI:** 10.3390/nu14183764

**Published:** 2022-09-13

**Authors:** Giulia Besutti, Massimo Pellegrini, Marta Ottone, Efrem Bonelli, Filippo Monelli, Roberto Farì, Jovana Milic, Giovanni Dolci, Tommaso Fasano, Simone Canovi, Stefania Costi, Stefania Fugazzaro, Marco Massari, Guido Ligabue, Stefania Croci, Carlo Salvarani, Pierpaolo Pattacini, Giovanni Guaraldi, Paolo Giorgi Rossi

**Affiliations:** 1Radiology Unit, Department of Diagnostic Imaging and Laboratory Medicine, Azienda USL–IRCCS di Reggio Emilia, 42123 Reggio Emilia, Italy; 2Department of Medical and Surgical Sciences, University of Modena and Reggio Emilia, 41124 Modena, Italy; 3Department of Biomedical, Metabolic and Neural Sciences, University of Modena and Reggio Emilia, 41124 Modena, Italy; 4Epidemiology Unit, Azienda USL–IRCCS di Reggio Emilia, 42123 Reggio Emilia, Italy; 5Clinical Chemistry and Endocrinology Laboratory, Azienda USL-IRCCS di Reggio Emilia, 42123 Reggio Emilia, Italy; 6Clinical and Experimental PhD Program, University of Reggio Emilia, 41124 Modena, Italy; 7Modena HIV Metabolic Clinic, University of Modena and Reggio Emilia, 41124 Modena, Italy; 8Scientific Directorate Azienda USL-IRCCS di Reggio Emilia, 42123 Reggio Emilia, Italy; 9Department of Surgery, Medicine, Dentistry and Morphological Sciences with Interest in Transplant, Oncology and Regenerative Medicine, University of Modena and Reggio Emilia, 41124 Modena, Italy; 10Physical Medicine and Rehabilitation Unit, Azienda USL–IRCCS di Reggio Emilia, 42123 Reggio Emilia, Italy; 11Infectious Diseases Unit, Azienda USL–IRCCS di Reggio Emilia, 42123 Reggio Emilia, Italy; 12Clinical Immunology, Allergy and Advanced Biotechnologies Unit, Azienda USL–IRCCS di Reggio Emilia, 42123 Reggio Emilia, Italy; 13Rheumatology Unit, Azienda USL–IRCCS di Reggio Emilia, 42123 Reggio Emilia, Italy

**Keywords:** COVID-19, follow-up, body composition, skeletal muscle, visceral fat

## Abstract

We aimed to describe body composition changes up to 6–7 months after severe COVID-19 and to evaluate their association with COVID-19 inflammatory burden, described by the integral of the C-reactive protein (CRP) curve. The pectoral muscle area (PMA) and density (PMD), liver-to-spleen (L/S) ratio, and total, visceral, and intermuscular adipose tissue areas (TAT, VAT, and IMAT) were measured at baseline (T0), 2–3 months (T1), and 6–7 months (T2) follow-up CT scans of severe COVID-19 pneumonia survivors. Among the 208 included patients (mean age 65.6 ± 11 years, 31.3% females), decreases in PMA [mean (95%CI) −1.11 (−1.72; −0.51) cm^2^] and in body fat areas were observed [−3.13 (−10.79; +4.52) cm^2^ for TAT], larger from T0 to T1 than from T1 to T2. PMD increased only from T1 to T2 [+3.07 (+2.08; +4.06) HU]. Mean decreases were more evident for VAT [−3.55 (−4.94; −2.17) cm^2^] and steatosis [L/S ratio increase +0.17 (+0.13; +0.20)] than for TAT. In multivariable models adjusted by age, sex, and baseline TAT, increasing the CRP interval was associated with greater PMA reductions, smaller PMD increases, and greater VAT and steatosis decreases, but it was not associated with TAT decreases. In conclusion, muscle loss and fat loss (more apparent in visceral compartments) continue until 6–7 months after COVID-19. The inflammatory burden is associated with skeletal muscle loss and visceral/liver fat loss.

## 1. Introduction

As of August 2022, there are almost 600 million COVID-19 survivors all over the world [1]. While most survivors have no clinical consequences after acute COVID-19, an unclear proportion of COVID-19 patients develop a “post-acute COVID-19 syndrome”, which has been differently defined over the course of the pandemic [2,3,4]. Recently, a WHO-led Delphi consensus defined this syndrome as a condition occurring usually three months after COVID-19 onset and with symptoms that last for at least two months and cannot be explained by an alternative diagnosis [5]. Symptoms are related to multiple organ systems, the most common being fatigue, shortness of breath, and cognitive dysfunction [5].

Baseline body composition influences short-term outcomes in infectious and inflammatory conditions including COVID-19, with sarcopenia and myosteatosis, obesity, and excess of ectopic/visceral adiposity being poor prognostic factors [6,7,8,9,10,11,12,13]. On the other hand, critical illnesses can influence body composition. Fat mass loss, but also muscle mass loss as a consequence of protein catabolism and prolonged immobilisation, are frequent in critically ill patients and may also be promoted by strong inflammatory reactions [14,15,16]. The recovery of this loss may take a long time, with data showing gain in fat mass being higher than gain in lean mass, resulting in a high proportion of sarcopenic patients 1 year after discharge [17,18].

Both baseline body composition and body composition modifications induced by COVID-19 may play a role in some of the manifestations of post-acute COVID-19 syndrome. For example, in a large analysis of 10 longitudinal studies and electronic health records in the UK, overweight and obese people were found to be at higher risk of long COVID [19,20].

A few studies have explored body composition changes after COVID-19. A low fat-free mass index was found in approximately 20% of patients three months after COVID-19 [21] while lean mass loss, abdominal fat mass loss, and an increase in liver density have been reported in intensive care unit (ICU) patients 20 days after admission [22]. Malnutrition and sarcopenia have been described in 60% to 87% of post-COVID-19 patients upon rehabilitation [23,24]. Greater weight loss and malnutrition three weeks after discharge were found in COVID-19 patients with higher CRPs and longer disease durations [25]. A decrease in body mass index (BMI) one month after COVID-19 has also been described, followed by BMI and waist circumference increasing from one to three months after COVID-19 in overweight/obese patients [26]. Finally, in a small study of 14 obese, mechanically ventilated COVID-19 survivors, an increase in fat mass and a decrease in lean mass were measured via bioelectrical impedance analysis between 3 and 6 months after ICU discharge [27]. However, these studies are limited by their short follow-up time, lack of baseline assessment, or selection bias (e.g., only ICU patients).

Hence, even if the effect of body composition on short-term COVID-19 outcomes is well-known, data on body composition changes after COVID-19 are scarce; therefore, their potential role in post-acute COVID-19 syndrome is still unclear.

The aims of this study are to describe changes in body composition parameters from diagnosis to three- and six-month follow-ups in severe COVID-19 survivors of the first pandemic wave in Italy, assessed through CT scan, and to evaluate the impact of the COVID-19 inflammatory burden on these changes.

## 2. Materials and Methods

### 2.1. Setting

In the Reggio Emilia province (Northern Italy, 532,000 inhabitants), the first case of SARS-CoV-2 infection was diagnosed on 27 February 2020. In Italy, the first wave of the pandemic lasted until May 2020. During this first wave, due to wild type SARS-CoV-2, most patients coming to the Emergency Department (ED) had severe disease, resulting in frequent hospitalizations, and the case fatality rate was high, reaching 20% in our region in the first weeks of the pandemic once a 30-day follow-up had been completed for all cases [28,29]. As of 15 May 2020, there had been 4863 RT-PCR-confirmed COVID-19 cases in the province. In the province, a “red-zone” lockdown was imposed from 8 March 2020 (with the exceptions for travel being proven work needs, emergencies, or health reasons); freedom of movement was re-established on 4 May and other non-essential activities re-opened later in the month.

In our institution, during the first wave of the pandemic, a baseline CT scan was routinely acquired for all patients with suspected COVID-19 pneumonia at the ED.

A routine 2–3-month follow-up CT scan was offered to all first-wave survivors who had been hospitalised during the disease course and had severe pneumonia with the following clinical-radiological features: respiratory failure during hospital stay (history of invasive or non-invasive mechanical ventilation and/or tocilizumab [TCZ] administration) or total extent of disease > 40% at baseline CT. A second CT scan at 6–7 months was proposed if residual CT abnormalities were present at 2–3 months [30].

In our institution, nutritional support during the hospital stay was provided only to mechanically ventilated patients by enteral tube feeding. The enteral tube feeding formula (Oxepa^®^-Abbott Nutrition) consisted of a complete, balanced, high-energy, low-carbohydrate, high-fat formula (energy 1,5 kcal/mL, protein 16.5%, carbohydrate 27.9%, fat 55.6%), enriched with eicosapentaenoic acid (EPA), gamma linolenic acid (GLA) and antioxidants.

### 2.2. Study Design and Ethics

This was a single-centre retrospective cohort study based on routinely collected data. The study was approved by the Area Vasta Emilia Nord (AVEN) Ethics Committee (protocol number 855/2020/OSS/AUSLRE, date of approval 28 July 2020). Given the retrospective nature of the data collection, the Ethics Committee authorises the use of a patient’s data without his/her informed consent if all reasonable efforts have been made to contact the patient to obtain it.

### 2.3. Study Population

All COVID-19 survivors who were offered a routine 2–3-month follow-up CT scan were considered eligible for the study. Patients who did not comply with the proposed follow-up protocol, patients without the second (6–7-month) follow-up CT scan, and patients with unsuitable CT scans for body composition measurements due to artifacts were excluded. Therefore, all consecutive COVID-19 patients who underwent baseline (T0) CT scan (between 27 February 2020 and 15 May 2020) and two follow-up CT scans 2–3 months (T1) and 6–7 months (T2) after COVID-19 pneumonia diagnosis were enrolled (Figure 1).

This study population is a subgroup of the previously reported study who described persistent lung CT abnormalities during follow-up and their association with the inflammatory burden during the disease course [30].

### 2.4. Data Collection

The COVID-19 Surveillance Registry, coordinated by the National Institute of Health and implemented in each Local Health Authority, was used to retrieve data on the dates of symptoms’ onset and diagnosis. Patients’ medical records were reviewed to collect data on comorbidities, COVID-19 treatment, and mechanical ventilation (invasive and non-invasive). The visually estimated extent of disease was retrieved from the structured reports of baseline CT scans [31]. All the available levels of CRP, D-dimer, and PaO2/FiO2 ratio during hospitalisation were collected. Measurement methods for CRP and D-dimer are reported have been previously reported [30]. Information on recent comorbidities was collected in the electronic medical records and through linkage with hospital discharge databases (2015–2019) in the previous 5 years, with the local Cancer Registry (period 2015–2019), and with the provincial diabetes registry, which included all prevalent cases of diabetes on 31 December 2019.

### 2.5. CT Retrospective Analysis

Acquisition parameters for the baseline and follow-up chest CT scans have been previously reported [30,31]. To evaluate the body composition parameters, both the baseline and follow-up CT images were retrospectively analysed by a single trained image analyser (EB) supervised by a senior radiologist (PP), both blinded to the clinical data and outcomes, by using the OSIRIX-Lite software V5.0 (Pixmeo, Sarl, Switzerland).

Methods for the measurement of chest CT body composition parameters have been previously reported [13,32,33,34].

Patients with CT scans not suitable for different post-processing evaluations were excluded from specific study analyses, e.g., CTs of patients with thoracic lipomas were excluded from the evaluation of fat compartments, and CTs with artifacts due to pacemakers or other implants were not suitable for pectoral muscle segmentation. When the field of view was not large enough to include all subcutaneous fat, the SAT was visually estimated.

### 2.6. Outcomes

The outcomes measured were body composition changes between the follow-up CT scan (at 2–3 and at 6–7 months) and the baseline CT scan (∆T1-T0 and ∆T2-T0), including changes in TAT, SAT, VAT, IMAT, L/S ratio, pectoral muscle area, and pectoral muscle density.

### 2.7. Putative Determinants

Putative determinants were age, sex, the interval of CRP curves during the disease course as a marker of inflammation, the highest D-dimer value during the disease course as a marker of endothelial activation, one of the major severe COVID-19 pathogenetic pathways, and length of hospital stay (LOS) as a generic index of disease severity and duration of immobilisation and dietary restraint. We also adjusted for baseline TAT (as a measure of general adiposity), as it necessarily influences most of the variations that can occur in body composition and fat compartments. Details of the methods used to compute the CRP interval have already been reported elsewhere [30]. Other known disease severity measures which could be involved in body composition changes, such as pneumonia extension at baseline, the need for mechanical ventilation (mostly determined by the lowest PaO2/FiO2 during the disease course), and steroid therapy were not considered as putative determinants because they were inclusion criteria or strictly linked to inclusion criteria for the study.

### 2.8. Data Analyses

We reported continuous variables as median (interquartile range—IQR) or mean (standard deviation—SD) and categorical variables as numbers and percentages (%). The changes of chest CT body composition parameters between different time points are reported as average values with 95% confidence intervals (95% CI), computed according to a normal distribution of the sample means.

The Spearman correlation was performed between putative determinants, especially between the CRP interval and LOS, to assess the potential presence of collinearity. For each of the CT body composition parameter changes between baseline and follow-up time points, we built multivariable linear regression models adjusted for age, sex, and baseline TAT, and we included the other putative determinants one by one; i.e., the standardised variables of CRP interval, D-dimer peak, and LOS. After categorising the baseline CT body composition parameters in tertiles, some sensitivity analyses were performed by restricting multivariable models to the last TAT and VAT tertiles, the first tertile of pectoral muscle area, and the first tertile of liver-to-spleen ratio, representing patients with sarcopenia and steatosis as pre-existing conditions.

The *p* values are reported as continuous measures and no significance threshold was set.

All statistical analyses were conducted using Stata/IC 16.1 statistical software (Stata Corp., College Station, TX, USA).

## 3. Results

### 3.1. Study Population

During the first wave of the pandemic, 1513 patients were hospitalised for COVID-19 in Reggio Emilia provincial hospitals. Of them, 1152 survived, and among the survivors, 351 individuals were invited to perform a routine follow-up CT scan at 2–3 months from diagnosis if they had experienced respiratory failure during the hospital stay or had a baseline CT disease extension ≥ 40%. Of those 351, 92 patients refused and 259 underwent a T1 follow-up CT scan. After excluding 18 patients who did not undergo a second follow-up CT scan at 6–7 months and 33 patients with unsuitable baselines, T1, or T2 follow-up CT scans due to artifacts, 208 COVID-19 survivors with three suitable time point CT scans were included in the study (Figure 1).

The mean age was 65.6 ± 11 years and 65 (31.3%) patients were female. The median baseline BMI was 29.1 kg/m^2^. Comorbidities and disease severity measures are reported in Table 1. The median hospital stay was 18 days; 100 (48.1%) patients required non-invasive and/or invasive mechanical ventilation.

### 3.2. Body Composition Changes

On average, a decrease in pectoral muscle area, TAT, VAT, and IMAT was observed both from baseline to 2–3 months after diagnosis and from 2–3 months to 6–7 months after diagnosis, although the decrease was greater in the first interval than in the second. Similarly, the increase in L/S ratio (i.e., a steatosis decrease) was greater in the first 2–3 months after diagnosis but continued up to 6–7 months (Figure 2). Pectoral muscle density, however, remained substantially stable in the first 2–3 months after diagnosis, and averagely increased (i.e., decreasing myosteatosis) from 2–3 to 6–7 months. Proportionally, fat decrease was greater in visceral fat compartments (VAT, L/S ratio) when compared to the SAT or TAT (Table 2 and Appendix A). BMI was available for only 136 patients at baseline and the T1 time point. From diagnosis to 2–3 months after COVID-19, a 1-point average decrease in BMI was registered (from mean BMI 29.8 ± 5.2 to mean BMI 28.8 ± 5.2). BMI was strongly associated with the TAT and SAT at the two time points (Appendix A). Figure 2 shows the trajectories over time for body composition parameters. For all fat compartments, including the liver and pectoral muscle area, an initial decrease followed by an increase was a common trajectory.

### 3.3. Impact of the Inflammatory Burden on Body Composition Changes

The effect of the CRP interval on body composition changes (∆T1-T0 and ∆T2-T0) was evaluated in models adjusted by age, sex, and baseline TAT (Figure 3).

Increasing the CRP interval was associated with a larger decrease in pectoral muscle area (linear regression coefficient of 1 SD increase in CRP interval for ∆T2-T0 = −0.78, 95%CI −1.41;−0.14, *p* = 0.017) and smaller increase in pectoral muscle density (linear regression coefficient of 1 SD increase in CRP interval for ∆T2-T0 = −1.55, 95%CI −2.73;−0.37, *p* = 0.010). For both parameters, the association was similar for ∆T1-T0.

Increasing the CRP interval was associated with a larger decrease in BMI 2–3 months after diagnosis (linear regression coefficient of 1 SD increase in the CRP interval for ∆T1-T0 = −0.965, 95%CI −1.25;−0.69, *p* < 0.001). As for CT-assessed body fat compartments, the direction of association was towards a greater decrease in fat compartments with an increasing CRP interval, but associations were compatible with random fluctuation for the TAT, SAT, and IMAT; a random fluctuation was not plausible only for the VAT (linear regression coefficient of 1 SD increase in the CRP interval for ∆T1-T0 = −1.67, 95%CI −2.85;−0.50, *p* = 0.005; linear regression coefficient of 1 SD increase in the CRP interval for ∆T2-T0 = −1.55, 95%CI −2.99;−0.11, *p* = 0.035) and the liver-to-spleen ratio when considering changes after 2–3 months (linear regression coefficient of 1 SD increase in the CRP interval for ∆T1-T0 = −0.073, 95%CI 0.038;0.107, *p* < 0.0001; linear regression coefficient of 1 SD increase in the CRP interval for ∆T2-T0 = 0.024, 95%CI −0.015;0.063, *p* = 0.23) (Figure 3).

Similar models were built for D-dimer (Figure 3), showing no associations. LOS was also tested as a putative determinant, but due to its collinearity with the CRP interval (linear regression coefficient = 0.39) and milder associations with body composition changes than those observed for the CRP interval, models were not considered informative.

Sensitivity analyses (Appendix A) showed milder associations of the CRP interval with body composition changes in patients with high TAT and in sarcopenic patients (lowest tertile of pectoral muscle area), with the exception of a higher association between the CRP interval and IMAT decrease in sarcopenic subjects. On the contrary, in patients with higher visceral fat and a lower liver-to-spleen ratio (patients with steatosis), the effect of the CRP interval on body composition changes was generally stronger than in the whole study population (for example, in patients in the highest VAT tertile, the linear regression coefficient of 1 SD increase in the CRP interval for pectoral muscle area ∆T1-T0 was −1.48, 95%CI −2.43;−0.52, *p* = 0.003).

## 4. Discussion

By evaluating serial CT scans of survivors of severe COVID-19 pneumonia, we observed a larger muscle and fat loss in the first 2–3 months after diagnosis, and a smaller but persistent loss 6–7 months after diagnosis. Visceral compartments including VAT and liver steatosis were more involved in fat loss than subcutaneous compartments. The inflammatory burden registered during COVID-19 was associated with muscle loss and visceral, liver, and muscle fat loss, while COVID-19-related endothelial activation was not associated with body composition changes.

Other studies have reported a rapid weight loss followed by a medium-term weight increase [26,27], similarly to what has been observed following other critical illnesses or acute respiratory distress syndrome [17,18]. In our study, even if the average muscle and fat loss decreased over time, we did not observe a recovery in muscle areas and in fat compartments, particularly in ectopic/visceral fat. The difference could be due to the severity of included patients: in the study by De Filippo et al., all survivors who had been hospitalised were eligible, but a low compliance to follow-up was observed [26]. In the present study, only severe cases–who needed Tocilizumab administration and/or mechanical ventilation or with more than 40% lung involvement at baseline CT–were eligible, and we obtained high compliance with the proposed follow-up. A lower muscle and fat loss in less severe cases (e.g., with a lower CRP interval and lower LOS) was also observed in our study, thus suggesting that in much less severe patients, an inverse trend could be possible after the first months. Furthermore, even if the average measures did not show a trend of restoring pre-disease levels, for most biomarkers, Figure 2 shows that part of the patients had a decrease in the first period followed by an increase in the second period.

Patients with a higher inflammatory burden are those who end up with the most unfavourable skeletal muscle characteristics, as they experience a higher loss in muscle area and a lower increase in muscle density. On the other hand, fat loss is higher in these patients, especially in visceral compartments. A possible explanation of the association between the inflammatory burden and muscle and fat loss lies in the direct association between inflammation and hospital stay duration. In fact, the association between CRP levels and LOS has already been reported [35,36]. However, the CRP interval showed stronger associations with body composition changes when compared to LOS, leading us to hypothesise that at least part of the effect of the inflammatory burden is independent of hospital stay duration. In a study of 213 COVID-19 survivors, both the CRP levels and LOS were associated with weight loss registered approximately three weeks after discharge, but in multivariable models, only LOS remained significantly associated with weight loss [25]. However, in this study, only CRP levels at admission and the CRP peak during hospitalisation were taken into account, while it is plausible that medium-term outcomes, particularly body composition-related outcomes that need time to change, depend on both intensity and duration of the inflammation that are better expressed by the CRP interval than by CRP values at one point.

Surprisingly, the study shows a different behaviour of visceral and inter-muscular adipose tissue compared to intra-muscular fat. In fact, unlike other fat deposits, pectoral density–which correlates with muscle lipid content [37,38]–increased only from T1 to T2 and increasing CRP interval was associated with a smaller PMD increase and a larger VAT decrease. This discrepancy with higher CRP levels being associated with increased myosteatosis but lower SAT and IMAT has already been reported in the general population [39], leading us to hypothesise that the inflammatory state may favour a smaller decrease of the PMD in COVID-19 survivors as well.

The main limitation of this study is that patient selection was based on disease severity, which makes it challenging or even impossible to infer valid associations for less severe patient cohorts. In fact, severity was not only a selection factor for patients to be included in the initial follow-up, but it also determined who should attend the second follow-up. Therefore, the body composition trajectories that were available for this study were only those of patients who had not completely recovered by the 2–3-month CT; fortunately, only a few patients had a completely negative CT at 2–3 months (7%). Moreover, the inclusion criteria of mechanical ventilation and baseline CT extension were strongly associated, leading to a collider effect that prevents us from evaluating the effect of these factors on body composition changes. Similarly, the study design–particularly, the need for anti-inflammatory drugs (TCZ) as an inclusion criterion–did not allow us to disentangle the interaction between disease-induced inflammation and therapies on the changes of body composition parameters. In fact, the inflammatory burden registered during the disease course (i.e., the CRP curve) was influenced not only by the disease, but also by anti-inflammatory drugs. According to the study design, our cohort comprised survivors only. In the interpretation of our results, it is important to consider that an important proportion of patients with similar baseline characteristics had died. It is also worth noting that the cohort was recruited during the first pandemic wave, when most patients were diagnosed at the ED with severe disease, were frequently hospitalized, and had a high fatality rate. These conditions did not apply to the subsequent pandemic waves due to the progressive improvement in the diagnostic process, the dissemination of vaccines, and the spread of the Omicron variant. Another peculiarity of the first COVID-19 wave in Italy was the concurrent lockdown, which, by inducing a reduction in physical activity and changes in dietary intakes, had the potential to contribute to body composition change. Data on the impact of lockdown are controversial: a 2021 systematic review and meta-analysis reported an increase in body weight and BMI in the post-lockdown period compared to the pre-lockdown period [40]; however, one study reported a significant weight loss in the elderly population, probably related to malnutrition and sarcopenia [41], and a recent study conducted in postmenopausal Spanish women reported no significant changes in fat and lean mass, measured by bioelectrical impedance analysis comparing the pre- and post-lockdown periods [42]. Still, we cannot exclude that the body composition changes that we registered were partially influenced by the lockdown, at least for patients discharged in March-April. Furthermore, the lack of height and BMI measurements in approximately one third of patients at baseline and for all patients at the 6–7-month follow-up prevented us from obtaining a measure of muscle quantity related to patient height and from evaluating the full BMI trajectories over time. Due to CT scan limits, the SAT was visually estimated in patients with higher subcutaneous adiposity, and thus the results on the SAT should be considered cautiously. Finally, the sample size was determined by the number of available patients and not by a formal power calculation. Nevertheless, the study was able to identify changes in body composition parameter of about 3% as statistically significant according to 95% CIs, which is a difference very close to, if not below, the minimum clinically significant difference [43,44]. On the other hand, subgroup analyses were underpowered, and our stratified results should be considered only as suggestive of differences in the body composition changes between groups.

Major strengths are the prospective data collection (notwithstanding the retrospective study design), the high patient compliance to the proposed follow-up protocol, and the evaluation of body composition by means of CT scans. In fact, even if the use of chest images is less validated and standardised than abdominal ones [33,45], it allowed us to obtain information on both the quantity and quality of skeletal muscle and on the distribution of body fat in different compartments. Thus, by analysing CT scans originally performed to evaluate lung parenchyma, comprehensive information on patients’ body composition was obtained without the need to expose the patients to adjunctive radiations.

## 5. Conclusions

In conclusion, body composition changes after COVID-19 include skeletal muscle loss and fat loss that are more apparent in visceral compartments. These losses are greater in the first months but continue up to 6–7 months after COVID-19, and they are greater in patients who experienced a higher inflammatory burden during the disease course. More studies are needed to explore if these body composition changes may be associated with functional impairment and symptoms reported by patients with post-acute COVID-19 syndrome and if this association is causal. If that is the case, assessment of body composition could be included as a step to individualise rehabilitation programmes for post-COVID-19 patients. Moreover, the present study opens a research question to be tested in a clinical trial regarding the possible clinical utility of targeted nutritional support during the COVID-19 disease course in patients with higher inflammatory burden, aimed at reducing the negative effects of COVID-19 on body composition.

## Figures and Tables

**Figure 1 nutrients-14-03764-f001:**
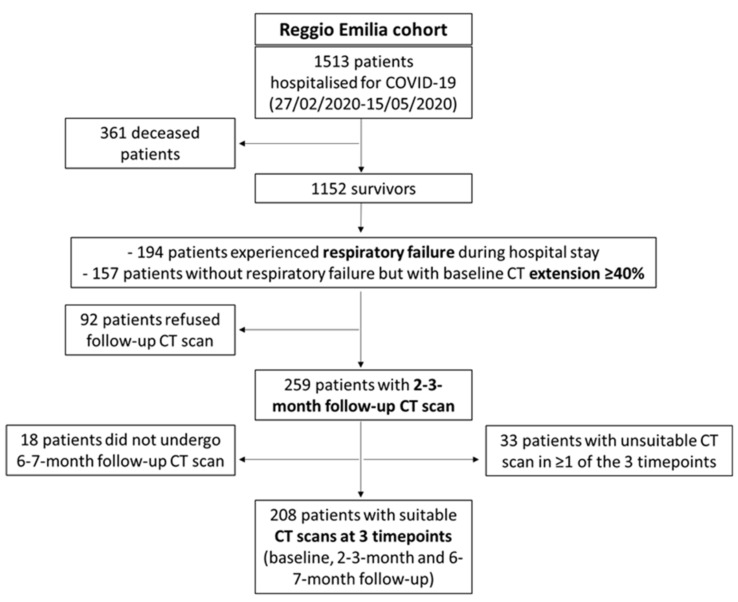
Patient inclusion and exclusion flow chart.

**Figure 2 nutrients-14-03764-f002:**
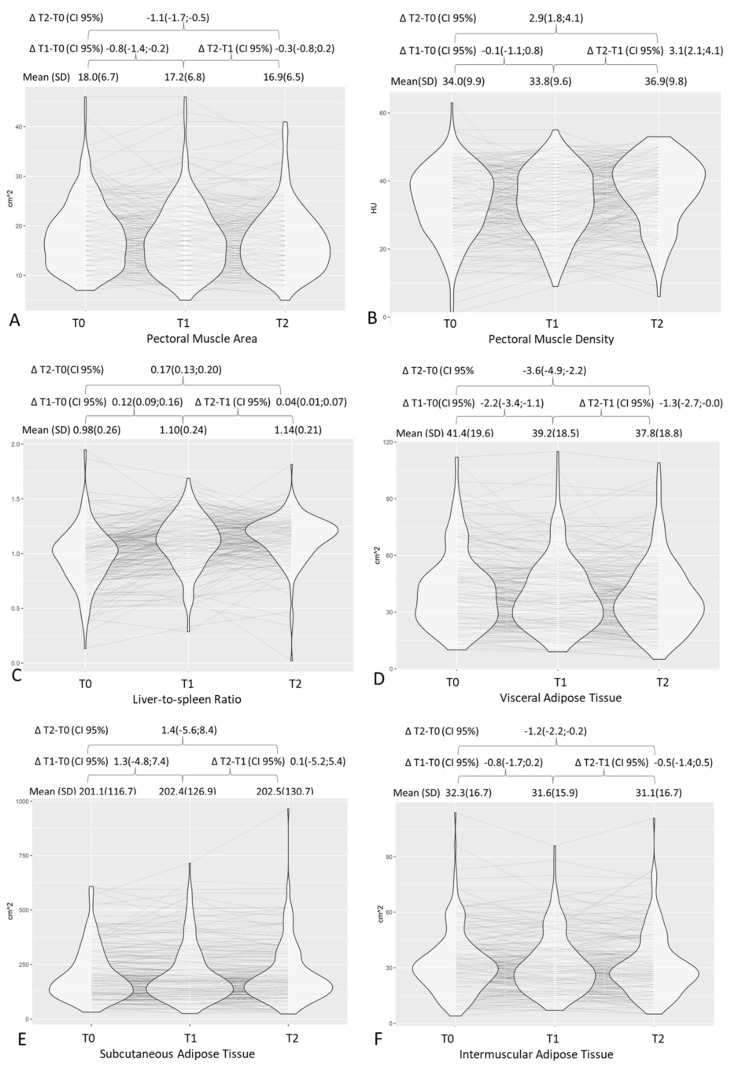
Distribution of chest CT body composition parameters at different time points with graphic representation of trajectories over time and respective mean values and deltas. T0, baseline; T1, 2–3 months after diagnosis; T2, 6–7 months after diagnosis. Graphs are for pectoral muscle area (**A**), pectoral muscle density (**B**), liver-to-spleen ratio (**C**), visceral adipose tissue (**D**), subcutaneous adipose tissue (**E**), and intermuscular adipose tissue (**F**).

**Figure 3 nutrients-14-03764-f003:**
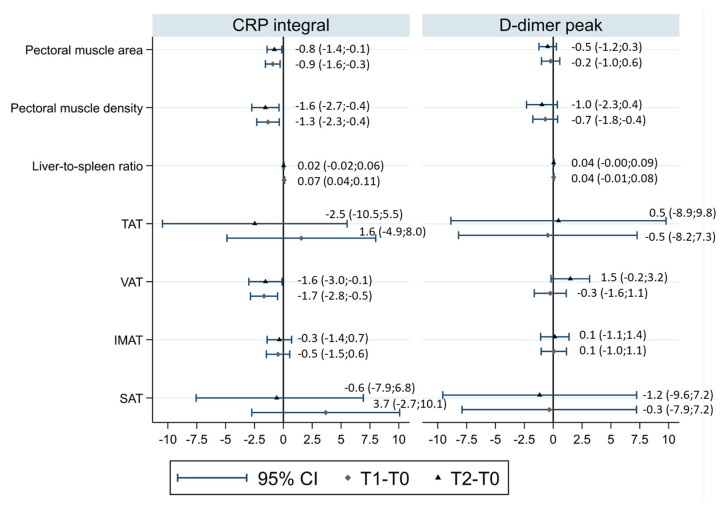
Associations of CRP intervals and D-dimer peaks with CT body composition changes in models adjusted for age, sex, and baseline TAT. Each diamond represents the linear regression coefficient (and the corresponding 95% confidence interval (CI)) of the CRP intervals and D-dimer peaks for CT body composition changes in T1–T0, and each triangle represents the linear regression coefficient of the CRP intervals and D-dimer peaks for CT body composition changes in T2–T0. TAT, total adipose tissue area; VAT, visceral adipose tissue area; SAT, subcutaneous adipose tissue area; IMAT, intermuscular adipose tissue area.

**Table 1 nutrients-14-03764-t001:** Baseline clinical characteristics and COVID-19 severity measures of the included patients.

		All Patients
		*n* (%)
		N = 208
Age (years), median (IQR)		65 (58–74)
Female sex		65 (31.3)
Smoking habit	Never	178 (85.6)
	Previous	27 (13.0)
	Current	3 (1.4)
COPD		8 (3.9)
Asthma		9 (4.3)
Cardiovascular diseases		52 (25.0)
Cancer		23 (11.1)
Diabetes		48 (23.1)
Hypertension		102 (49)
Chronic kidney failure		3 (1.4)
Cerebrovascular disease		16 (7.7)
Liver diseases		6 (2.9)
Baseline BMI (kg/m^2^) *		29.1 (26.1–33.1)
Days from symptom onset, median (IQR)		6 (4–9)
Baseline CT disease extension		
0		1 (0.5)
<20		12 (5.8)
20–40		51 (24.5)
40–60		107 (51.4)
≥60		37 (17.8)
CRP interval, median (IQR)		168 (96–253)
Tocilizumab administration **		56 (53.3)
Steroid therapy		61 (29.3)
Non-invasive mechanical ventilation		86 (41.3)
Invasive mechanical ventilation		26 (24.1)
Mechanical ventilation		100 (48.1)
Lowest PaO_2_/FiO_2_ (mmHg), median (IQR) ***		120 (81–219)
D-dimer peak (ng/mL), median (IQR) ****		1816 (926–4236)
Length of hospital stay (days), median (IQR)		18 (12–29)

BMI, Body Mass Index; COPD, Chronic Obstructive Pulmonary Disease; IQR, Interquartile range. Missing values: * 72 (34.6%) for baseline BMI; ** 103 (49.5%) for Tocilizumab administration; *** 13 (6.3%) for Lowest PaO_2_/FiO_2_; **** 51 (24.5%) for D-dimer peak.

**Table 2 nutrients-14-03764-t002:** Chest CT body composition parameters at each time point and respective changes between different time points.

	Mean (SD)			Mean Difference (95%CI)		
	T0	T1	T2	∆T1-T0	∆T2-T1	∆T2-T0
Pectoral Muscle Area (*n* = 202)	18.01 (6.69)	17.23 (6.81)	16.90 (6.48)	−0.78 (−1.39;−0.17)	−0.33 (−0.84;0.18)	−1.11 (−1.72;−0.51)
Pectoral Muscle Density (*n* = 203)	33.97 (9.94)	33.82 (9.61)	36.89 (9.76)	−0.15 (−1.09;0.79)	3.07 (2.08;4.06)	2.92 (1.79;4.05)
Liver-to-spleen Ratio (*n* = 184)	0.98 (0.26)	1.10 (0.24)	1.14 (0.21)	0.13 (0.09;0.16)	0.04 (0.01;0.07)	0.17 (0.13;0.20)
IMAT (*n* = 204)	32.3 (16.7)	31.6 (15.9)	31.1 (16.7)	−0.75 (−1.72;0.23)	−0.47 (−1.45;0.52)	−1.21 (−2.23;−0.19)
VAT (*n* = 205)	41.4 (19.6)	39.2 (18.5)	37.8 (18.8)	−2.21 (−3.35;−1.08)	−1.34 (−2.67;−0.01)	−3.55 (−4.94;−2.17)
TAT (*n* = 203)	274.8 (133.2)	273.1 (142.9)	271.7 (148.2)	−1.67 (−7.89;4.55)	−1.46 (−7.18;4.26)	−3.13 (−10.79;4.52)
SAT (*n* = 203)	201.1 (116.7)	202.4 (126.9)	202.5 (130.7)	1.33 (−4.77;7.43)	0.09 (−5.23;5.41)	1.42 (−5.57;8.41)

T0, baseline; T1, 2–3 months after diagnosis; T2, 6–7 months after diagnosis. SD, standard deviation; TAT, total adipose tissue area; VAT, visceral adipose tissue area; SAT, subcutaneous adipose tissue area; IMAT, intermuscular adipose tissue area.

## Data Availability

Participant data that underlie the results reported in this manuscript will be shared after de-identification, beginning 6 months and ending at least 7 years after article publication, to researchers who provide a methodologically sound proposal with objectives consistent with those of the original study. Proposals and data access requests should be directed to the Area Vasta Emilia Nord (AVEN) Ethics Committee at CEReggioemilia@ausl.re.it as well as to the Authors at the Epidemiology Unit of AUSL–IRCCS di Reggio Emilia at info.epi@ausl.re.it, who are the data guardians. To gain access, data requestors will need to sign a data access agreement.

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
