# Peer review of "Modifications of Chest CT Body Composition Parameters at Three and Six Months after Severe COVID-19 Pneumonia: A Retrospective Cohort Study"

_nutrients, 2022, doi:10.3390/nu14183764_

Round 1
Reviewer 1 Report
This study includes a valuable purpose what focused relationship between COVID-19 pneumonia and body composition, and the results may contribute to the development of pandemic control and public health.
However, there are serious problems, the manuscript needs to make a significant correction for the reviewing and proofreading of it. In particular, the following points should be revised.
1. Changes in body composition can be attributed to both the infection of COVID-19 and changes in living environment due to the pandemic (such as lockdown). In this study, is it possible to separately examine these two effects? If not, it is necessary to state the cautionary note of the interpretation in more detail than the discussion part.
2. Figures 2 and 3 are too small to read, please modify the layout and font size.
3. Coronaviruses have showed different symptoms depending on the type. Please indicate the type of coronavirus (Delta, Omicron, etc.).
4. In the introduction, please explain the characteristics of COVID-19 pandemic in Italy (e.g., how it differs from other countries). Moreover, please discuss the impact of these points on the results in this study.
5. After revising the above points, check the STROBE guideline for cohort study. Please submit the checklist. (https://www.strobe-statement.org/checklists/).
Author Response
RE: We thank the Reviewer for the comments that allowed us to improve our work.
- Changes in body composition can be attributed to both the infection of COVID-19 and changes in living environment due to the pandemic (such as lockdown). In this study, is it possible to separately examine these two effects? If not, it is necessary to state the cautionary note of the interpretation in more detail than the discussion part.
RE: In this study, it is not possible to separately estimate the effect of COVID-19 and related inflammatory burden and the effect of dietary and physical activity changes induced by the lockdown on body composition changes in COVID-19 survivors. We added in the method section (“setting”) a sentence on the lockdown in our region: “In the province, a “red-zone” lockdown was imposed from 8 March 2020 (with exceptions for travels being proven work needs, emergencies, or health reasons); freedom of movements was re-established on 4 May and other not essential activities re-opened later in the month.” In the discussion section, among limitations induced by having conducted the study during the first pandemic wave, a new paragraph has been added: “Another peculiarity of the first COVID-19 wave in Italy was the concurrent lockdown, that by inducing a reduction in physical activity and changes in dietary intakes had the potential to contribute to body composition change. Data on the impact of lockdown are controversial: a 2021 systematic review and meta-analysis reported an increase in body weight and BMI in the post-lockdown compared to the before lockdown period [40], however one study reported a significant weight loss in the elderly population, probably related to malnutrition and sarcopenia [41], and a recent study conducted in postmenopausal Spanish women reported no significant changes in fat and lean mass measured by bioelectrical impedance analysis comparing the pre- and post-lockdown periods [42]. Still, we cannot exclude that the body composition changes that we registered were partially influenced by the lockdown, at least for patients discharged in March-April.”
- Bakaloudi DR, Barazzoni R, Bischoff SC, Breda J, Wickramasinghe K, Chourdakis M. Impact of the First COVID-19 Lockdown on Body Weight: A Combined Systematic Review and a Meta-Analysis. Clin Nutr. 2021; S0261-5614(21)00207-7, doi: 10.1016/j.clnu.2021.04.015.
- Di Santo SG, Franchini F, Filiputti B, Martone A, Sannino S. The Effects of COVID-19 and Quarantine Measures on the Lifestyles and Mental Health of People Over 60 at Increased Risk of Dementia. Front Psychiatry. 2020; 11:578628, doi: 10.3389/fpsyt.2020.578628.
- Acedo C, Roncero-Martín R, Sánchez-Fernández A, et al. Body Composition and Nutrients Dietary Intake Changes during COVID-19 Lockdown in Spanish Healthy Postmenopausal Women. Eur J Investig Health Psychol Educ. 2022; 12:631–638. doi: 10.3390/ejihpe12060047.
- Figures 2 and 3 are too small to read, please modify the layout and font size.
RE: We agree with the Reviewer, Figures have been changed accordingly.
- Coronaviruses have showed different symptoms depending on the type. Please indicate the type of coronavirus (Delta, Omicron, etc.).
RE: Since the included patients were the first to be infected and hospitalized in one of the first provinces affected by the first wave of the pandemic, as explained in the section methods – setting (and now even more stressed as it is reported also in the aim) this was the first Sars-COV-2 that affected European countries (wild type), and not one of the subsequent variants. This has been clarified in methods-setting.
- In the introduction, please explain the characteristics of COVID-19 pandemic in Italy (e.g., how it differs from other countries). Moreover, please discuss the impact of these points on the results in this study.
RE: In the introduction – aim, we clarified that the included patients were from the first pandemic wave (“The aims of this study are to describe changes in body composition parameters from diagnosis to three- and six-month follow-up in severe COVID-19 survivors of the first pandemic wave in Italy, assessed through CT scan, and to evaluate the impact of COVID-19 inflammatory burden on these changes.”). As to the characteristics of the first wave in Italy, in order to leave the focus of the introduction on body composition, we preferred to explain them in Methods – Setting, where we have added: “During this first wave due to wild type SARS-CoV-2, most patients presenting at the Emergency Department (ED) had severe disease, resulting in frequent hospitalizations, and the case fatality rate was high, reaching 20% in our region in the first weeks of the pandemic once 30-day follow-up had been completed for all cases [28,29].” In the discussion – limitation section we have already stated some consequences of the peculiarities of the first wave: “According to the study design, our cohort is constituted of survivors only. In the interpretation of our results, it is important to consider that an important proportion of patients with similar baseline characteristics has died. It is also worth noting that the cohort was recruited during the first pandemic wave when most patients were diagnosed at the ED with severe disease, were frequently hospitalized and there was a high fatality rate. These conditions do not apply to the subsequent pandemic waves, due to the progressive improvement in the diagnostic process, the dissemination of vaccines and the spread of the Omicron variant.”
- Giorgi Rossi P, Broccoli S, Angelini P, et al; Emilia-Romagna COVID-19 working group. Case fatality rate in patients with COVID-19 infection and its relationship with length of follow up. J Clin Virol. 2020;128:104415. doi:10.1016/j.jcv.2020.104415.
- Giorgi Rossi P, Marino M, Formisano D, et al; Reggio Emilia COVID-19 Working Group. Characteristics and outcomes of a cohort of COVID-19 patients in the Province of Reggio Emilia, Italy. PLoS One. 2020 Aug 27;15(8):e0238281. doi: 10.1371/journal.pone.0238281.
- After revising the above points, check the STROBE guideline for cohort study. Please submit the checklist. (https://www.strobe-statement.org/checklists/).
RE: We have filled the checklist. We have added “a retrospective cohort study” to the title.

Reviewer 2 Report
Giulia Besutti et al have developed a very interesting study illustrating the effect of COVID-19 infection. Overall the manuscript is well presented following the STROBE guidelines. I also consider that the statistical analysis has been correctly performed. However, there are a few questions and minor suggestions regarding this manuscript:
-The tables could be modified to make them easier to understand. I think the "Missing" column should be deleted (Table 1). In Table 2 "Mean (95%CI)" should be changed to "Mean difference (95%CI)".
-A comparison with patients without COVID-19 or with other infections would be interesting. I think this is impossible to do if the authors did not do this at the beginning of the study. However this can be discussed in the discussion section.
Author Response
Giulia Besutti et al have developed a very interesting study illustrating the effect of COVID-19 infection. Overall the manuscript is well presented following the STROBE guidelines. I also consider that the statistical analysis has been correctly performed. However, there are a few questions and minor suggestions regarding this manuscript:
-The tables could be modified to make them easier to understand. I think the "Missing" column should be deleted (Table 1). In Table 2 "Mean (95%CI)" should be changed to "Mean difference (95%CI)".
RE: We thank the Reviewer for the suggestions. We have changed the tables accordingly. Missing values in Table 1 have been added as footnotes.
-A comparison with patients without COVID-19 or with other infections would be interesting. I think this is impossible to do if the authors did not do this at the beginning of the study. However this can be discussed in the discussion section.
RE: We thank the Reviewer for the comment. Unfortunately, since body composition changes in our study have been measured through repeated CT scans, it was not possible to include a control group of negative patients or patients with other infections (repeated CT scans in other infections are not usually requested, unless indicated for specific clinical reasons). This would have also helped to disentangle the potential effect of lockdown (see Reviewer 1 comment). In the introduction, we added some points on body composition changes in critically ill patients, with the whole paragraph being now: “On the other hand, critical illnesses can influence body composition. Fat mass loss, but also muscle mass loss as a consequence of protein catabolism and prolonged immobilisation, are frequent in critically ill patients and may also be promoted by strong inflammatory reactions [14-16,15]. The recovery of this loss may need a long time, with data showing gain in fat mass being higher than gain in lean mass, resulting in a high proportion of sarcopenic patients 1 year after discharge [17,18].” Also in the discussion we have added a short reference to similar studies (“Other studies have reported a rapid weight loss followed by a medium-term weight increase [26,27], similarly to what has been observed following other critical illnesses or acute respiratory distress syndrome [17,18].).
- Joskova V, Patkova A, Havel E, et al. Critical evaluation of muscle mass loss as a prognostic marker of morbidity in critically ill patients and methods for its determination. J Rehabil Med. 2018 Aug 22;50(8):696-704. doi: 10.2340/16501977-2368.
- Lambell KJ, Goh GS; Tierney AC, et al. Marked Losses of Computed Tomography-Derived Skeletal Muscle Area and Density over the First Month of a Critical Illness Are Not Associated with Energy and Protein Delivery. Nutrition. 2021; 82:111061, doi: 10.1016/j.nut.2020.111061.
- Piotrowicz K, GÄ…sowski J, Michel JP, Veronese N. Post-COVID-19 acute sarcopenia: physiopathology and management. Aging Clin Exp Res. 2021 Oct;33(10):2887-2898. doi: 10.1007/s40520-021-01942-8.
- Thackeray M, Kotowicz MA, Pasco JA, Mohebbi M, Orford N. Changes in Body Composition in the Year Following Critical Illness: A Case-Control Study. J Crit Care 2022; 71:154043, doi: 10.1016/j.jcrc.2022.154043.
- Chan KS, Mourtzakis M, Aronson Friedman L, Dinglas VD, Hough CL, Ely EW, Morris PE, Hopkins RO, Needham DM; National Institutes of Health National Heart, Lung, and Blood Institute (NHLBI) Acute Respiratory Distress Syn-drome (ARDS) Network. Evaluating Muscle Mass in Survivors of Acute Respiratory Distress Syndrome: A 1-Year Mul-ticenter Longitudinal Study. Crit Care Med. 2018 Aug;46(8):1238-1246. doi: 10.1097/CCM.0000000000003183.
Finally, by conducting the literature search on other infection, we also found another small study on post COVID-19 body composition changes, which we have now included in the introduction and reference list. “Finally, in a small study of 14 obese mechanically ventilated COVID-19 survivors, an increase in fat mass and a decrease in lean mass were measured via bioelectrical impedance analysis between 3 and 6 months after ICU discharge [27].”
- Joris M, Minguet P, Colson C, et al. Cardiopulmonary Exercise Testing in Critically Ill Coronavirus Disease 2019 Survi-vors: Evidence of a Sustained Exercise Intolerance and Hypermetabolism. Crit Care Explor. 2021 Jul 13;3(7):e0491. doi: 10.1097/CCE.0000000000000491.
